# The burden of out-of-pocket and indirect costs of cutaneous leishmaniasis patients in Minas Gerais, Brazil

Sarah Nascimento Silva[1]*, Endi Lanza Galvão[1,2], Janaína de Pina Carvalho[1], Mayra Soares Moreira[1], Tália Santana Machado de Assis[1,3], Glaucia Cota[1]

1 Pesquisa Clínica e Políticas Públicas em Doenças Infecto-Parasitárias, Núcleo de Avaliação de Tecnologias em Saúde, Instituto René Rachou, Fundação Oswaldo Cruz, Belo Horizonte, Brazil, 2 Programa de Pós-Graduação em Reabilitação e Desempenho Funcional, Universidade Federal dos Vales do Jequitinhonha e Mucuri, Diamantina, Brazil, 3 Centro Federal de Educação Tecnológica de Minas Gerais, Contagem, Brazil

* sarah.nascimento@fiocruz.br

## Abstract

### Background

Healthcare expenses represent a proportionally greater burden for the poorest populations, which further exacerbates the negative impact of the disease on the individual's quality of life and productivity.

### Objective

The study aimed to identify the direct and indirect costs during the treatment of cutaneous leishmaniasis (CL) from the patients' perspective and examine factors influencing the costs burden among CL patients.

### Methods

A prospective cost analysis was conducted between April 2022 and April 2023 through interviews with patients with a confirmed diagnosis of CL. Direct costs were estimated using the micro-costing approach, and indirect costs using the human capital method. Descriptive analyses and hypothesis tests were conducted for associations between costs and sociodemographic and clinical variables, with a significance level of 5%.

### Results

The study included 68 patients, predominantly male (77.9%) with an average age of 53 years. Cutaneous leishmaniasis was the most common clinical form (76.4%), with new cases accounting for 79.4% of participants. Patients averaged 3.5 outpatient visits per CL treatment cycle, with miltefosine and intravenous meglumine antimoniate being the most prescribed therapies. Direct costs per treatment cycle averaged

**Data availability statement:** The authors confirm that all data underlying the findings are fully available without restriction Our data is available at Mendeley Data: https://data.mendeley.com/datasets/xhmmnch5nv/1.

**Funding:** This study was supported by Fundação Oswaldo Cruz (Grant: VPPCB-008-FIO-18-2-80 - Programa Inova Fiocruz to SNS), and Conselho Nacional de Desenvolvimento Científico e Tecnológico - CNPq (Grant:302069/2022-4 to GC) The funders had no role in study design, data collection and analysis, decision to publish, or preparation of the manuscript.

**Competing interests:** The authors have declared that no competing interests exist.

USD 117.36, attributed to transportation, food, and medical exams. Indirect costs from lost workdays amounted to USD 9,936.58, with an average of USD 160.12 per patient. Catastrophic expenditure (>10% of monthly income) was observed in 42.6% of families, significantly associated with direct cost, bacterial infection, and sociodemographic factors such as gender, age, and distance traveled.

## Conclusions

This study underscores the substantial economic burden of CL treatment on patients, highlighting the need for targeted interventions to mitigate financial hardship, particularly among vulnerable socioeconomic groups.

## Autor summary

Cutaneous leishmaniasis (CL) is a neglected disease that mainly affects the most vulnerable populations in tropical and subtropical countries. The disease causes lesions on the skin and mucous membranes that can lead to physical effects, functional impairment and loss of productivity, as well as psychological and economic impacts on individuals. Although the disease is common in some regions, there are few treatment options, many of which are toxic and require monitoring. Many patients are referred to specialised services that receive patients from different regions to diagnose and treat the disease. In this study, patients at a referral centre were asked about the out-of-pocket expenses they have to pay while receiving treatment for CL that are not covered by the health system, such as transport, food, medicines, services and absence from work. The results of these studies will be important in measuring costs from the patient's perspective and understanding the impact that costs during treatment for CL can have on family income.

## Introduction

The composition of treatment costs is a crucial piece of information in the context of neglected diseases, as it is closely associated with conditions of social vulnerability and limited access to healthcare resources [1]. Healthcare expenses represent a proportionally greater burden for the poorest populations [2], which further exacerbates the negative impact of the disease on the individual's quality of life and productivity [3]. With occurrences intrinsically linked to regions with low levels of social and economic development, diseases in the neglected group perpetuate a vicious cycle, being both a result of and a contributing factor to poverty [4,5].

Cutaneous leishmaniasis (CL) is a neglected and endemic disease in several states in Brazil. The skin and mucosal wounds caused by the disease can hinder the performance of daily tasks and routines. In severe cases, destructive lesions in the mucous membranes of the upper airways can impair vital functions [6]. In Brazil,

the Unified Health System (SUS) provides treatment for leishmaniasis through a strategic, comprehensive care program. This program includes consultations, diagnostic and therapeutic procedures, hospitalizations, prevention and rehabilitation actions, and the provision of standardized medications as outlined in the Clinical Guidelines and Protocols [6]. The treatment regimens for CL in Brazil include six therapeutic options and are indicated according to clinical criteria and the characteristics of the lesion, the duration of which can vary between 10 and 30 days [6]. Thus, the direct medical costs of CL treatment, from the health system's perspective [7], are anticipated and universally covered by the SUS service network. These costs are included in a fixed budget for strategic health programs under the Ministry of Health [8].

To fully estimate the costs involved in CL treatment, it is important to consider the patient's perspective, more specifically "out-of-pocket health expenditures", which has been relatively underexplored [7]. In addition to direct medical costs for procedures, exams, consultations, and medications, patients may also incur expenses for travel, food, accommodation, and other non-medical expenses generated by the treatment. Along side direct costs, indirect costs—such as those related to the impact on the individual's quality of life and loss of productivity—must also be considered [7,9].

Studies dedicated to measuring the expenses for the treatment of leishmaniasis from the perspective of patients in Brazil are still scarce, with analyses from the perspective of the health system prevailing [10–12]. Even in countries with a public and universal health system, such as Brazil, a third of households spend more than 10% of their budget on health, with a more significant impact on the poorest families [13]. An analysis by the World Bank shows that 27% of health spending is paid out of pocket in Brazil, although 20% is the recommended amount [14]. Understanding the spending profile of these patients and identifying strategies to minimize these costs [15], whether through funding assistance actions or policies to promote equity, is fundamental to confronting, minimizing the impact of, and eradicating the disease in the country. This study aimed to identify the direct and indirect costs during the treatment of CL from the patients' perspective and examine factors influencing the costs burden among CL patients.

## Methods

### Ethics statement

Ethical approval for the study was granted by the Ethics Committee on Human Research of the René Rachou Institute, Fundação Oswaldo Cruz (CAAE protocol # 28929220.0.0000.5091; approval number 3.918.626, March 16, 2020). The participants were patients over 12 years old with confirmed cutaneous or mucosal leishmaniasis, who were consecutively enrolled after signing the informed consent form between April 2022 and April 2023. Written informed consent was obtained from the parents or legal guardian of participants under 18 years old using Informed Consent Form (ICF), and Assent Form (AF) was obtained from the children prior to participation. Patients who had difficulties understanding the established criteria (some illiterate patients, deaf and dumb patients, frail elderly patients, disabled patients) or who did not agree to respond to recruitment approaches were excluded from the study.

This is a cost analysis conducted from the perspective of patients, with data obtained from a prospective study carried out at the leishmaniasis referral center, René Rachou Intitute, Fundação Oswaldo Cruz, in Belo Horizonte, Minas Gerais, Brazil.

Data were collected through individual interviews conducted by four trained researchers, utilizing a semi-structured questionnaire (S1 File) supplemented by reviews of medical records once during the treatment. A treatment for cutaneous leishmaniasis ranges from 10 to 30 days depending on each therapeutic regimen. The data collection period for treatment-related costs was set as the time between the initial treatment dose and 30 days after the final dose, corresponding to one complete cycle of CL treatment. The expenditures detailed in this study pertain to a single treatment cycle for patients and do not encompass additional cycles initiated due to treatment failures, disease recurrences, or new cases in the same patient.

The sociodemographic data were collected using a standardized form developed by the Brazilian Association of Research Companies (ABEP) of 2021, which categorizes individuals into five socioeconomic strata according to average

household income: A (USD 4,698.99; BRL 22,749.24), B1 (USD 2,228.44; BRL 10,788.56), B2 (USD 1,181.86; BRL 5,721.72), C1 (USD 659.81; BRL 3,194.33), C2 (USD 391.41; BRL 1,894.95), and DE (USD 178.14; BRL 862.41). Disease characteristics, such as clinical presentation, number of lesions, treatment regimen, comorbidities, type of cases (initial episode or recurrence), presence of bacterial infections associated with leishmaniasis, and occurrence of adverse events during treatment, were extracted from medical records.

Researchers devised a semi-structured form to document direct medical and non-medical expenses incurred during a cycle of CL treatment, encompassing costs for transportation, meals, lodging, medical consultations and exams, dressings, and medications reported by patients and/or their companions. The number of consultations reported in interviews reflects the patient's clinical monitoring, excluding visits to other healthcare facilities such as hospitals, health centers, or primary health care assessments. Indirect costs related to lost workdays and hours due to CL treatment were also assessed, as well as potential costs associated with caregivers.

Direct non-medical and indirect costs were evaluated across six therapeutic options: intralesional meglumine antimoniate infiltration, parenteral meglumine antimoniate (with or without pentoxifylline), miltefosine, fluconazole, and liposomal amphotericin B. The selection of the treatment performed, which considered the patient's clinical profile, medical history, and relevant factors discussed with the patient, was not influenced by the present study.

Costs were estimated using a micro-costing approach from the patient's perspective. Direct costs for each category (medical and non-medical) were aggregated based on reported amounts, with recurring costs per visit calculated according to the number of visits documented in the patient's medical records. Indirect costs were assessed using the Human Capital Method (HCM), considering the number of missed workdays (absenteeism) and the reported remuneration by patients. The HMC considers the number of missed workdays multiplied by the daily wage. The hours the patient reported having missed from their everyday activities or the approximate amount of time travelled to the CRL-IRR were taken into consideration, according to either their account or estimate value of appointments and distance traveled (<20km-2h; 50–75km – 4h; over 75km or transportation subsidized by the Health Departments of the patient's city – 8h). The calculus also considered the number of appointments and hospitalization time of each patient. Costs related to adverse effects were estimated based on self-reported information collected during the patient interviews. Participants who experienced any adverse effect associated with treatment were asked to report all related out-of-pocket expenses, including spending on medications, medical consultations, transportation, or other costs incurred specifically for the management of those effects. These values were included in the total cost estimates per patient when applicable.

The Brazilian currency (Real: R$) was used for cost calculations, standardized in terms of minimum monthly salaries (MMS) set by the Brazilian government. Income impairment was calculated by dividing the expenditure amount by the patient's income, and expenditures exceeding 10% of the family income were indicative of catastrophic expenses [16]. We converted the monetary value of each patient's expenses from Brazilian Reais (BRL) into US dollars (USD) at the exchange rate on the 29st of December, 2023 (USD 1.00 = R$ 4,8413).

A descriptive statistical analysis was conducted, presenting data with measures such as mean, standard deviation, minimum and maximum values, median, and relative and absolute frequencies. For non-normally distributed continuous variables assessed using the Shapiro-Wilk test, the Mann-Whitney test was employed to investigate the association between reported expenditures on the disease and the occurrence of catastrophic expenditures. A significance level of 5% was applied to all statistical tests.

## Results

The study population comprised 68 patients, 53 (77.9%) were male at an average age of 53 years (±17). The most prevalent clinical form of the disease was cutaneous, affecting 52 (76.4%) individuals, while mucosal and mucocutaneous forms affected 8 (11.7%) individuals each. Among those interviewed, 79.4% (n = 54) represented new cases of CL. On average, patients visited the outpatient department (referral center) 3.5 times (median 3, IQR 3–4) during one cycle of CL

treatment. Among the therapeutic options used, miltefosine was the most prescribed (n = 24; 35.3%), followed by meglumine antimoniate with intravenous administration (n = 20; 29.4%) (Table 1).

Most patients reported engaging in paid work (n = 40; 58.8%) or relying on retirement benefits or government assistance as their source of income (n = 22; 32.3). Only 8.8% (n = 6) of patients reported having no income. The majority of participants (76.5%) had a monthly income of up to USD 500.69 (2 MMS), with a median income of USD 250.35 (1 MMS). The predominant economic classes were C (45.6%) and DE (36.8%). Adding direct medical and non-medical costs, the total average cost of one cycle of CL treatment per patient was USD 117.36 (USD 0.00- USD 1,157.75) with 50% of patients reporting a median of up to USD 58.04 (IQR 1–3; 31.24 – 135.81). The most significant expenses were on transportation, food, and medical exams (Fig 1), ranging from USD 28.54 (0.00-185.90) and USD 25.19 (0.00-154.92), to USD 24.82 (0,00-330.49), respectively (Table 2). Half of the patients made use of transportation subsidized by the Health Departments of the patient's city of residence, as defined in the SUS legislation.

Indirect costs arising from lost workdays (absenteeism) due to CL treatment amounted to USD 9,936.59 (n = 62). The average value of the indirect cost was USD 160.12 (0.00 – 801.11) per patient (Table 2). These costs encompassed the period during which the patient underwent medication use, consultations at the referral center, and received care from other ancillary services, including hospitalization within one cycle of CL treatment. The need for a companion during consultations related to CL was reported by 21 (30.9%) participants, most of whom (n = 35; 57.4%) reported losing a day of work due to the treatment. In this study, 11 (16.1%) of the patients who were hospitalized to get treatment for an average of 16 days (±7.7) contributed to 45.8% (USD 3,123.07) of the total indirect cost.

Despite the distinct treatment cycles and dosages associated with CL using miltefosine, IV meglumine antimoniate, intralesional meglumine antimoniate, and liposomal amphotericin B, the average expenditure per treatment from the patient's perspective exhibited minor variations in both direct costs (USD 114.74 – 120.54) and indirect costs (USD 144.95 - 148.16). Similarities were also noted in the number of consultations at the referral center, travel distances, and the average percentage of family income allocated to treatment among patients using these therapies. However, notable discrepancies were observed for treatments involving fluconazole or IV meglumine antimoniate + pentoxifylline, with participation limited to only one participant (Table 3).

The analysis of monthly income allocation for CL treatment revealed that 36 patients (52.9%) expended more than 10% of their individual monthly income on treatment, with 22 of them exceeding 25% of their income. Catastrophic expenditure was observed in 42.6% of families (26 of 61 families, for 7 family members there was not enough data for this calculation). Among clinical variables, catastrophic expenditure was significantly associated solely with bacterial infection (p = 0.021), and not with variables such as number of lesions, comorbidities, recurrence, treatment type, number of consultations, adverse events, or hospitalization requirement. Sociodemographic factors linked to catastrophic expenditure included gender (p < 0.001), age (p = 0.046), and distance traveled (p = 0.018).

Furthermore, there was a significant association between total direct costs and catastrophic expenditure (p < 0.001), whereas expenses related to medical services and other costly sources showed no such association (Table 4).

## Discussion

This study identified that, despite the availability of CL therapy through Brazil's SUS, there is a significant economic burden on patients due to treatment costs. More than two-thirds of the study participants earn up to two minimum wages, making the treatment costs a substantial portion of their monthly income. Direct treatment expenses were found to contribute to household catastrophic spending. Transportation and food emerged as primary expenditures for patients, with over 42% of them exceeding 10% of their family monthly income. Indirect treatment costs revealed frequent work absenteeism, with hospitalization significantly increasing the societal burden of leishmaniasis treatment.

Family health expenditures are presently a metric monitored by the WHO Global Health Observatory to detect catastrophic out-of-pocket health spending. This term refers to health expenditures that surpass families' financial capacities,

**Table 1. Sociodemographic and clinical characteristics of patients with cutaneous leishmaniasis attended at the Leishmaniasis Reference Center, Minas Gerais, 2022 a 2023 (n=68).**

| Variables | n | % | Average cost (DP) | MMS* |
|---|---|---|---|---|
| **Gender** | | | | |
| Male | 53 | 77.9 | 108.21 (127.94) | 0.43 |
| Female | 15 | 22.1 | 149.70 (288.66) | 0.60 |
| **Age** (years) | | | | |
| 15-29 | 9 | 13.2 | 73.56 (86.83) | 0.29 |
| 30 a 59 | 32 | 47.1 | 132.01 (207.31) | 0.53 |
| ≥60 | 27 | 39.8 | 114.61 (154.54) | 0.46 |
| **Region of origin** | | | | |
| Metropolitan region of Belo Horizonte | 34 | 50.0 | 113.92 (197.47) | 0.46 |
| Country side of the state of Minas Gerais | 34 | 50.0 | 120.81 (150.78) | 0.48 |
| **Education level*** | | | | |
| Illiterate | 2 | 2.9 | 38.73 (13.87) | 0.15 |
| Middle School | 31 | 45.6 | 128.63 (208.83) | 0.51 |
| High school | 13 | 19.1 | 99.28 (112.69) | 0.40 |
| Higher education | 14 | 20.6 | 69.88 (81.43) | 0.28 |
| NR | 8 | | | |
| **Carries out paid work** | | | | |
| Yes | 40 | 58.8 | 87.41 (95.48) | 0.35 |
| No | 28 | 41.2 | 160.16 (243.05) | 0.64 |
| **Place of residence** | | | | |
| Belo Horizonte (BH) and Metropolitan Region of BH | 34 | 50 | 113.92 (197.47) | 0.46 |
| Countryside of Minas Gerais | 34 | 50 | 120.81 (150.78) | 0.48 |
| **Distance traveled** | | | | |
| <49km | 31 | 45.6 | 113.67 (206.24) | 0.45 |
| 50Km or more | 37 | 54.4 | 120.46 (145.39) | 0.48 |
| **Transportation** | | | | |
| City subsidy | 34 | 50 | 108.85 (135.64) | 0.43 |
| No subsidy | 34 | 50 | 125.88 (207.86) | 0.50 |
| **Clinical form of leishmaniasis** | | | | |
| Cutaneous (localized and spread) | 52 | 76.4 | 107.77 (130.09) | 0.43 |
| Mucosal | 8 | 11.7 | 185.00 (394.83) | 0.74 |
| Both cutaneous and mucosal | 8 | 11.7 | 112.10 (77.32) | 0.45 |
| **Associated comorbidities*** | | | | |
| Yes | 44 | 64.7 | 129.86 (199.53) | 0.52 |
| No | 23 | 33.8 | 97.16 (117.22) | 0.39 |
| NR | 1 | | | |
| **Number of lesions*** | | | | |
| Up to 3 lesions | 48 | 70.6 | 81.66 (81.22) | 0.33 |
| 4 or more | 13 | 19.1 | 105.43 (89.93) | 0.42 |
| NR | 7 | | | |
| **Ulcerated Lesion*** | | | | |
| Yes | 55 | 80.8 | 115.10 (186.96) | 0.46 |
| No | 12 | 17.6 | 126.93 (111.33) | 0.51 |
| NR | 1 | | | |

*(Continued)*

**Table 1.** (Continued)

| Variables | n | % | Average cost (DP) | MMS* |
|---|---|---|---|---|
| **Bacterial infection** | | | | |
| Yes | 10 | 14.7 | 217.40 (341.50) | 0.87 |
| No | 52 | 76.5 | 100.11 (123.61) | 0.40 |
| **New case of leishmaniasis** | | | | |
| Yes | 54 | 79.4 | 109.84 (129.35) | 0.44 |
| No | 14 | 20.6 | 146.39 (295.63) | 0.58 |
| **Treatment** | | | | |
| IV meglumine antimoniate | 20 | 29.4 | 86.68 (76.51) | 0.35 |
| Intralesional meglumine antimoniate | 12 | 17.6 | 71.08 (103.77) | 0.28 |
| Miltefosine | 24 | 35.3 | 145.81 (228.54) | 0.58 |
| Lipossomal Anfotericin B | 10 | 14.7 | 145.58 (224.65) | 0.58 |
| Fluconazole | 1 | 1.5 | 46.48 (0) | 0.19 |
| IV meglumine antimoniate + Pentoxifylline | 1 | 1.5 | 392.46 (0) | 1.57 |
| **Account of adverse events** | | | | |
| Yes | 21 | 30.9 | 121.02 (178.87) | 0.48 |
| No | 47 | 69.1 | 109.17 (167.92) | 0.44 |
| **Number of appointments** | | | | |
| Up to 3 | 42 | 61.8 | 141.96 (213.73) | 0.57 |
| 4 or more | 26 | 38.2 | 77.62 (62.73) | 0.31 |
| **Hospitalization** | | | | |
| Yes | 11 | 16.2 | 237.54 (372.23) | 0.95 |
| No | 57 | 83.8 | 94.16 (90.89) | 0.38 |

NR, not related.

*In Brazilian, the minimum monthly salaries (MMS), at that time was equivalent to US$ 250.35.

potentially leading to financial hardships [16]. This indicator is defined as health expenditures that constitute a significant portion, exceeding 10% or 25% of total household expenditure or income. In Brazil, from 2010 to 2019, there was a relative increase in the share of household spending allocated to health goods and services, reaching an average of 8% of available family income by 2019 [17]. The findings of this study suggest that the median expenditure on CL treatment alone reached catastrophic levels for over 40% of families, thereby imposing a financial burden on household budgets in addition to other healthcare expenses. These insights underscore the relevance of monitoring health expenditure as a proportion of household income to identify financial vulnerabilities.

A similar study conducted at the same service between 2015 and 2017, spanning a six-year interval, revealed that income inequalities persisted within this population [18]. However, it is noteworthy that expenditures on medication, previously identified as a significant cost component and reflective of catastrophic spending among the population of Minas Gerais, exhibited a notable reduction in this study [19]. This decline in medication expenses may reflect advancements in comprehensive pharmaceutical assistance policies, initiated since 2011 [20], alongside improvements in service organization in Minas Gerais in recent years [21,22].

While underexplored in economic studies [23], out-of-pocket expenses can pose significant barriers to accessing healthcare treatments. In neglected diseases, such expenses can markedly diminish the quality of life and financial resilience of low-income families across various countries [24]. Costs related to transportation and food, which constitute a substantial portion of direct expenses, underscore the critical need for improved access to healthcare facilities or enhanced transportation assistance through public management. For instance, a study examining the expenditures of

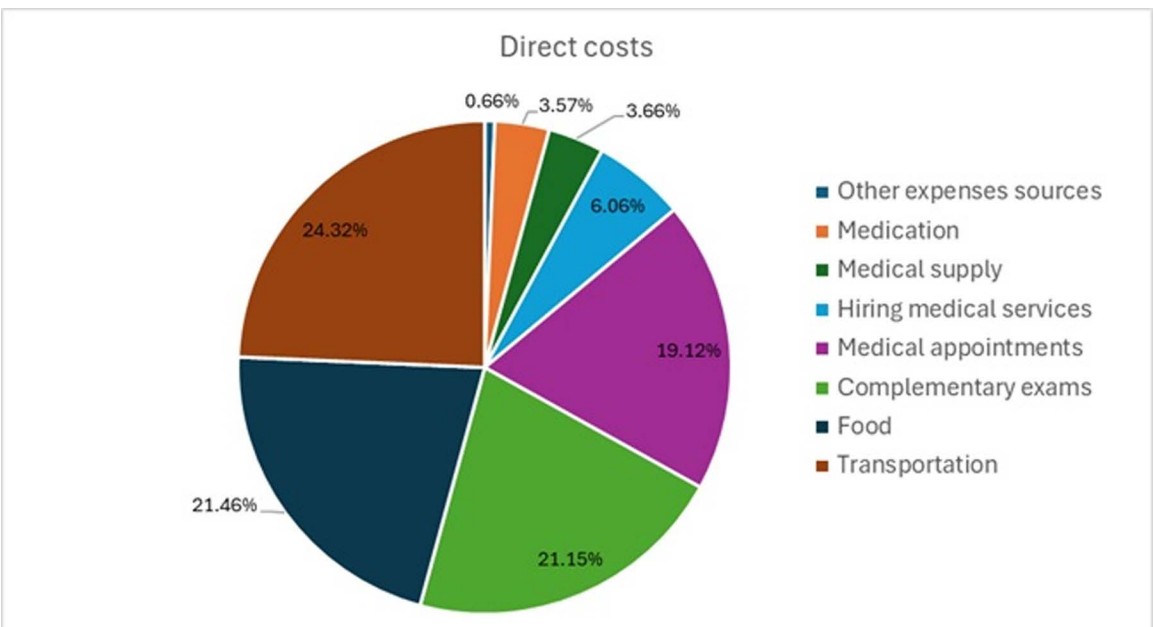

**Fig 1. Patients' expenses during CL treatment by category, Minas Gerais, 2022 a 2023.**

**Table 2. Income and expenses patterns and direct medical and non-medical costs from the perspective of patients with cutaneous leishmaniasis, Minas Gerais, 2022 a 2023 (n = 68).**

|  | Mean | ±SD | Median | IQR (Q1-Q3) | Maximum | Mean in MMS* |
|---|---|---|---|---|---|---|
| **Patient's monthly income** | 582.97 | 736.56 | 250.35 | 250.35 - 500.69 | 4,593.81 | 2.33 |
| **Per capita income** | 450.73 | 690.75 | 500.69 | 125.17 - 500.69 | 4,593.81 | 1.8 |
| **Family income** | 1.079.66 | 1.047.36 | 500.69 | 500.69 – 1,752.42 | 5,006.92 | 4.31 |
| **Direct cost** |  |  |  |  |  |  |
| Transportation | 28.54 | 37.43 | 12.39 | 0 - 40.07 | 185.90 | 0.11 |
| Food | 25.19 | 32.46 | 15.49 | 0 - 33.05 | 154.92 | 0.1 |
| Medication | 4.20 | 10.63 | 0.00 | 0 - 0.77 | 61.97 | 0.02 |
| Medical supply | 4.30 | 12.17 | 0.00 | 0 - 2.43 | 82.62 | 0.02 |
| Hiring medical services | 7.11 | 37.78 | 0.00 | 0.00 – 0.00 | 274.72 | 0.03 |
| Medical appointments | 22.44 | 90.06 | 0.00 | 0.00 – 0.00 | 619.67 | 0.09 |
| Complementary exams | 24.82 | 61.48 | 0.00 | 0.00 – 0.00 | 330.49 | 0.1 |
| Other expense sources | 0.77 | 5.01 | 0.00 | 0.00 – 0.00 | 41.31 | 0 |
| **Total direct cost** | 117.36 | 173.11 | 58.04 | 31.24 - 135.81 | 1.157.75 | 0.47 |
| **Indirect cost** | 160.12 | 169.60 | 106.40 | 40.68 - 204.97 | 801.11 | 0.64 |
| **Total cost (direct + indirect)** | 266.97 | 255.21 | 175.61 | 97.27 - 326.48 | 1.295.44 | 0.22 |

SD, standard deviation; IQR, interquartile range.

*In Brazilian, the minimum monthly salaries (MMS), at that time was equivalent to US$ 250.35.

patients undergoing treatment for Chagas disease explored this aspect: patients receiving care at a specialized referral hospital incurred five times higher transportation costs, experienced three times greater income loss, and spent four times longer commuting compared to those treated at local primary care facilities [25]. Higher expenditure was also associated

**Table 3. Treatment costs and patient impacts in leishmaniasis therapy, Minas Gerais, 2022 a 2023 (n = 68).**

| Treatment | N | Treatment time in days (mean) | Number of consultations (mean) | Distance traveled (KM mean) | Direct cost (mean) | Indirect cost (mean) | Family income (mean) | Family income spent on CL treatment (%) | Catastrofic spending (>10% Family income spent) | Direct cost in MMS* |
|---|---|---|---|---|---|---|---|---|---|---|
| IV meglumine antimoniate | 20 | 24.3 | 3.56 | 176.9 | 119.95 | 147.14 | 1.078.13 | 11.13 | yes | 0.59 |
| Intralesional meglumine antimoniate | 12 | 27.2 | 3.55 | 170.33 | 114.74 | 148.16 | 1.090.44 | 10.52 | yes | 0.59 |
| Miltefosine | 24 | 28.3 | 3.54 | 163.94 | 117.14 | 144.95 | 1.193.47 | 9.82 | No | 0.58 |
| Lipossomal Anfotericin B | 10 | 20 | 3.56 | 164.82 | 120.54 | 147.62 | 1.081.78 | 11.14 | Yes | 0.59 |
| Fluconazole# | 1 | 30 | 2 | 691.59 | 46.48 | 25.03 | 1.001.38 | 4.64 | No | 0.1 |
| IV meglumine antimoniate + Pentoxifylline# | 1 | 30 | 4 | 308.2 | 392.46 | 143.05 | 1.001.38 | 39.19 | yes | 0.57 |

KM, kilometres.

*In Brazilian, the minimum monthly salaries (MMS), at that time was equivalent to US$ 250.35.

#Only one patient (n = 1).

**Table 4. Factors associated with catastrophic family expenditure during cutaneous leishmaniasis treatment, Minas Gerais, 2022 a 2023 (n = 61).**

| Direct cost | Catastrophic spending | | |
|---|---|---|---|
| | Yes | No | p-valor |
| | | (Mean ±SD) (USD) | (Mean ±SD) (USD) |
| Transportation | 38.83 ± 45.16 | 18.98 ± 22.32 | 0.09* |
| Food | 42.18 ± 39.58 | 12.89 ± 18.83 | 0.001* |
| Medication | 8.44 ± 15.35 | 1.23 ± 5.11 | 0.002* |
| Dressing materials | 8.27 ± 18.47 | 0.79 ± 1.61 | 0.013* |
| Hiring medical services | 16.13 ± 59.87 | 1.82 ± 10.82 | 0.371 |
| Medical appointments | 42.06 ± 130.61 | 1.43 ± 6.40 | 0.016* |
| Complementary exams | 42.26 ± 66.67 | 7.37 ± 29.80 | 0.001* |
| Other expense sources | 1.87 ± 8.11 | 0.12 ± 0.70 | 0.179 |
| Total direct cost** | 200.05 ± 221.43 | 44.67 ± 42.31 | <0.001* |

*p < 0.05

**Cost and income data reported for only 61 patients.

with CL treatment for patients who travelled longer distances in the study carried out between 2015–2017 at the same center [18]. In this study, even though half of the patients received transportation subsidies, this variable corresponded to the largest component of direct non-medical spending and compromised approximately 10% of the MMS in force in 2023.

The association between catastrophic family expenditure with the direct cost of CL treatment indicates that these expenses constitute a substantial non-covered medical expense to most patients, reinforcing findings about the impact of CL on vulnerable populations [26]. The occurrence of out-of-pocket costs, mainly on transportation and food, were also the main components of health expenses in other studies [18,27]. Although the definition of CL treatment is based on clinical data and characteristics of the wounds, there are situations in which the discussion of therapeutic options may consider patient preferences and other factors related to therapy [28].

Understanding patients' profiles, which income and spending patterns during treatment, provides relevant data for discussing access to services and identifying factors that may adversely affect treatment adherence or continuity.

Furthermore, alongside the challenges of planning and delivering services in a regionalized manner [29], the evaluation of strategic policies such as that of CL must encompass crucial aspects to ensure universal and equitable access to healthcare, incorporating the patient's perspective. Several strategies documented in the literature aim to reduce out-of-pocket health expenditures, including regulatory measures, resource creation, financing, and service delivery [30]. Additionally, it is important to consider aspects that elucidate the impact of reducing out-of-pocket expenses according to socioeconomic status [31–33].

Absenteeism resulting from CL treatment adds another layer of economic burden for patients and their families. It is important to highlight that although CL is in general not a chronic but a prolonged disease (months). The varying effectiveness of certain treatments and the development of the mucosal form can lead to complications, which can require additional treatment cycles. The total indirect costs incurred during a treatment cycle, as depicted in this study, underscore CL's significant impact on patients' ability to work, exacerbated for those requiring hospitalization. Hospitalization not only entails high medical expenses [11] but also results in prolonged productivity loss, contributing to nearly half of the observed indirect costs. This finding emphasizes the necessity for public health policies to ensure improved access to appropriate and safe outpatient treatments.

Our study identified patient-reported out-of-pocket expenditure during referral centre interviews in a small and limited sample of patients collected prospectively over 12 months. Some patients required additional services when opting for therapeutic alternatives like amphotericin or IV meglumine. Although we asked patients about all expenses throughout their treatment, some costs may have been underreported or omitted due to contextual factors, memory bias, interviewee discomfort, and financial constraints. Another notable bias was the influence of estimated treatment costs on patients' treatment choices, highlighting the need for improved cost modeling in future studies. Additionally, the study's findings are constrained by the local health system's organization and the specific characteristics of the population studied, limiting the generalizability of conclusions. Nonetheless, these results shed light on significant aspects of CL treatment in Brazil, offering a patient-focused perspective often overlooked in healthcare decision-making.

This study emphasizes the patient demographics and economic challenges associated with CL treatment, advocating for policies that address practices, access, and patient support. The findings stress the need for a comprehensive approach to CL treatment that encompasses both clinical considerations and the socioeconomic impact on patients. Investing in prevention, early diagnosis, and cost-effective, accessible treatments can reduce the financial burden on patients and healthcare systems alike. Therefore, further studies confirming this patient profile are crucial to inform shared decision-making, promote access, equity, and adherence to CL treatments.

## Supporting information

**S1 File. Semi-structured questionnaire.**
(DOCX)

## Acknowledgments

Instituto René Rachou, Fundação Oswaldo Cruz for all support to develop the research, Programa Institucional de Bolsas para Iniciação Científica of Fundação Oswaldo Cruz (PIBIC-Fiocruz) and Fundação de Amparo à Pesquisa de Minas Gerais (FAPEMIG) to support students during this project.

## Author contributions

**Conceptualization:** Sarah Nascimento Silva, Janaína de Pina Carvalho, Gláucia Cota.

**Data curation:** Sarah Nascimento Silva, Janaína de Pina Carvalho.

**Formal analysis:** Sarah Nascimento Silva, Endi Lanza Galvão, Janaína de Pina Carvalho, Mayra Soares Moreira, Tália Santana Machado de Assis.

**Methodology:** Sarah Nascimento Silva, Endi Lanza Galvão, Janaína de Pina Carvalho, Mayra Soares Moreira, Tália Santana Machado de Assis, Gláucia Cota.

**Project administration:** Sarah Nascimento Silva, Janaína de Pina Carvalho.

**Supervision:** Sarah Nascimento Silva.

**Writing – original draft:** Sarah Nascimento Silva, Endi Lanza Galvão, Gláucia Cota.

**Writing – review & editing:** Sarah Nascimento Silva, Endi Lanza Galvão, Janaína de Pina Carvalho, Mayra Soares Moreira, Tália Santana Machado de Assis, Gláucia Cota.

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
