## [Decision Letter · Decision Letter 0]

18 Mar 2025

Thank you for submitting your manuscript to PLOS Neglected Tropical Diseases. After careful consideration, we feel that it has merit but does not fully meet PLOS Neglected Tropical Diseases's publication criteria as it currently stands. Therefore, we invite you to submit a revised version of the manuscript that addresses the points raised during the review process.

Response to Reviewers
Revised Manuscript with Track Changes
Manuscript

Shaden Kamhawi

co-Editor-in-Chief

Paul Brindley

co-Editor-in-Chief

**Additional Editor Comments:**
**Journal Requirements:**

1) Please provide an Author Summary. This should appear in your manuscript between the Abstract (if applicable) and the Introduction, and should be 150-200 words long. The aim should be to make your findings accessible to a wide audience that includes both scientists and non-scientists. Sample summaries can be found on our website under Submission Guidelines:

2) Tables should not be uploaded as individual files. Please remove these files and include the Tables in your manuscript file as editable, cell-based objects. For more information about how to format tables, see our guidelines:

https://journals.plos.org/plosntds/s/tables 

3) In the online submission form, you indicated that "The authors will make their datasets available on request". All PLOS journals now require all data underlying the findings described in their manuscript to be freely available to other researchers, either

- In a public repository

- Within the manuscript itself

- Uploaded as supplementary information.

4) Please ensure that the funders and grant numbers match between the Financial Disclosure field and the Funding Information tab in your submission form. Note that the funders must be provided in the same order in both places as well. State the initials, alongside each funding source, of each author to receive each grant. For example: "This work was supported by the National Institutes of Health (####### to AM; ###### to CJ) and the National Science Foundation (###### to AM).".

**Reviewers' comments:**

**Key Review Criteria Required for Acceptance?**

**Methods:**

-Are the objectives of the study clearly articulated with a clear testable hypothesis stated?

-Is the study design appropriate to address the stated objectives?

-Is the population clearly described and appropriate for the hypothesis being tested?

-Is the sample size sufficient to ensure adequate power to address the hypothesis being tested?

-Were correct statistical analysis used to support conclusions?

-Are there concerns about ethical or regulatory requirements being met?

Reviewer #1: The study is well designed and conducted. Patient's characteristics are well described. It is important to understand how the authors reached the sample size, it seems to be low and underpowered. Therefore I ask the authors to explain it and consider that it may be a possible limitation to the study. Statistical analysis seems correct, no concerns about ethical or regulatory requirements.

Please explain better "indirect costs" and how it was measured.

Reviewer #2: I think the objectives of the study are appropriate and important to study. Understanding the out-of-pocket costs, including the indirect costs for patients of CL will be useful from a policy perspective as well, particularly if it might be used for advocacy to generate further supports from government.

i think the population sampled were appropriate (and I assume limited by budget), though a multi-location design may have been more useful to exclude location specific cost effects and provide more generalizability to the results.

I have a question about why the data collection window appears to be between the first dose and 30 days after last dose. It was not clear whether the researchers go back to the same patient multiple times in that data collection period, or did they go only once? This is important to clarify, because if you are assessing costs only once at the beginning of the treatment, you may miss costs that occur later in the treatment (for example, adverse effects.) I appreciate that this is complicated slightly by the different duration of treatments for the different regimens.

The methods used otherwise appear to be well justified and appropriate, but I did have some questions about the analysis and the patients who were excluded from the study, which can be found in my comments in the editorial section.

The sample size itself was not very large (68), and there was only a single data point for some treatment regimens. In addition, getting data from multiple locations would have strengthened the results. But as a prospective study, this is fine. If possible, getting data from more locations, and for the different treatment regimens, would enable the running of more sophisticated analysis on this data set, including various regressions to control for and identify the most significant factors affecting the costs.

**Results:**

-Does the analysis presented match the analysis plan?

-Are the results clearly and completely presented?

-Are the figures (Tables, Images) of sufficient quality for clarity?

Reviewer #1: The data is well presented but it would be important to show:

1. In Figure 1 or table 2: specific direct costs versus type of treatment.

2. In table 4: catastrophic family expenditure versus type of treatment.

Reviewer #2: The small n in this case, restricts the analysis they could do here. Nevertheless, given the sample size, the results are well presented and clear to me.

There are some things that are interesting: for example, that ulcerated lesions have lower average costs than non-ulcerated lesions, but that may be due to the small sample size of non-ulcerated lesions.

Otherwise, the results, tables, and figures were clear to me.

**Conclusions:**

-Are the conclusions supported by the data presented?

-Are the limitations of analysis clearly described?

-Do the authors discuss how these data can be helpful to advance our understanding of the topic under study?

-Is public health relevance addressed?

Reviewer #1: Data are well discussed and the conclusions are well supported. Limitation due to sample size should be recognized.

Public relevance is well addressed and the paper will contribute to health care decision-making.

Reviewer #2: The discussion was appropriate and relevant. This would be good data for policy makers to see. I had some questions about things added to the discussion that were not apparent in the tables, but otherwise it looked good.

The limitations should note the geographic specificity of the data and also the relatively small sample size, which limited more advanced analysis.

**Editorial and Data Presentation Modifications?**

Reviewer #1: It is an important paper to be published, after minor revision modifications.

Reviewer #2: Please can you address the following questions:

1.) In the introduction section, please describe the typical diagnostic and treatment protocol in Brazil, so that it is easier to understand the duration of the diagnosis / treatment and understand what costs you are collecting? For example, if you are collecting cost data immediately after the first dose, you may miss adverse effects from the treatment that may appear later in the treatment cycle.

2,) Did the researchers go back to the same patient multiple times in that data collection period to collect cost data, or did they go only once? If they only surveyed a particular patient once for the costs, wouldn't the costs assessed after a first dose be different from the costs assessed after 20 doses? How would that affect your results?

3.) How do the results change between the over and under 18 age groups? I suggest you separate the analysis for the over and under 18 age-groups, simply because the direct and indirect costs will likely be different for these two groups.

4.) Were there location specific factors (since all the data was from one testing centre), that may have influenced the results?

5.) It is not clear what you mean when you say that "patients who had difficulty understanding the established criteria" were excluded from the study. Please explain this a little more? What criteria did they not understand?

6.) Do you have any sociodemographic data about the excluded patients (those who decided not to participate)? For example, were they disproportionately female, or of a certain ethnicity or income group? If so, that limitation should be noted during the discussion.

7.) How did you calculate the cost of adverse effects? Please clarify in the Methods section.

8.) Why did you not collect information on comorbidities like hypertension and diabetes, or other infectious diseases? The presence of comorbidities would likely have increased costs, correct? Is there a reason you didn’t you collect that data?

9.) Why does Table 4 only report for 61 patients? Is that because 61 out of the 68 patients experienced “catastrophic spending”? Please explain in the document and please relate this to line 278 where you note that “the median expenditure on CL treatment alone reached catastrophic levels for over 40% of families”. Moreover, where in the analysis that you have presented can we see that 40%?

10.) Please add the structured questionnaire that the authors used in the appendix and any details of how costs were calculated for each cost component? It would be useful for other researchers to see what exactly were the cost components that were measured, and how they were measured.

**Summary and General Comments:**

Reviewer #1: Measuring the costs of CL treatment in Brazil is a very important task. It is clear that the study will provide valuable data for health care decision-making, considering patient's needs and CL economic burden.

Reviewer #2: I thought this was an interesting paper which might be useful for advocacy around generating additional government support for CL patients. The sample size was not particularly large and it was geographically specific, but the results were still interesting.

PLOS authors have the option to publish the peer review history of their article (what does this mean? ). If published, this will include your full peer review and any attached files.

**Do you want your identity to be public for this peer review?** For information about this choice, including consent withdrawal, please see our Privacy Policy .

Reviewer #1: No

Reviewer #2: No

**Figure resubmission:****Reproducibility:** To enhance the reproducibility of your results, we recommend that authors of applicable studies deposit laboratory protocols in protocols.io, where a protocol can be assigned its own identifier (DOI) such that it can be cited independently in the future. Additionally, PLOS ONE offers an option to publish peer-reviewed clinical study protocols. Read more information on sharing protocols at https://plos.org/protocols?utm_medium=editorial-email&utm_source=authorletters&utm_campaign=protocols

---

## [Editor Report · Decision Letter 1]

1 Apr 2025

Dear PhD Silva,

We are pleased to inform you that your manuscript 'The burden of out-of-pocket and indirect costs of cutaneous leishmaniasis patients in Minas Gerais, Brazil' has been provisionally accepted for publication in PLOS Neglected Tropical Diseases.

Best regards,

Helen P Price, PhD

Academic Editor

Hira Nakhasi

Section Editor

Shaden Kamhawi

co-Editor-in-Chief

Paul Brindley

co-Editor-in-Chief

Thank you to the authors for their clear and comprehensive responses to the comments from the reviewers. I am satisfied that the manuscript is now suitable for publication.

---

## [Editor Report · Acceptance letter]

Dear PhD Silva,

We are delighted to inform you that your manuscript, "The burden of out-of-pocket and indirect costs of cutaneous leishmaniasis patients in Minas Gerais, Brazil," has been formally accepted for publication in PLOS Neglected Tropical Diseases.

Best regards,

Shaden Kamhawi

co-Editor-in-Chief

Paul Brindley

co-Editor-in-Chief
